# Generalized Linear Models for Describing Tree Gaps in Forest Management Areas in the Brazilian Amazon

**Suzana Ligoski Zeferino** [1],*, **Natally Celestino Gama** [1],*[ID], **Deivison Venicio Souza** [1][ID], **Alex Soares de Souza** [2], **Emil José Hernández Ruz** [1][ID] and **Sandra Dezuite Balieiro da Silva** [3]

1   Graduate Program in Biodiversity and Conservation, Federal University of Pará-UFPA, Altamira University Campus, Altamira 68370-000, Pará, Brazil; deivisonvs@ufpa.br (D.V.S.); emilhjh@ufpa.br (E.J.H.R.)
2   Secretaria de Estado do Meio Ambiente e Sustentabilidade do Pará (SEMAS), Belém 66093-677, Pará, Brazil; alex_souza3@hotmail.com
3   LN Guerra Indústria e Comércio de Madeiras LTDA, Belém 66814-385, Pará, Brazil; sandra.balieiro@lnguerra.com.br
*   Correspondence: sligoskizeferino@gmail.com (S.L.Z.); natallygama28@gmail.com (N.C.G.)

**Abstract:** Gap size is one of the main variables used to quantify the environmental consequences of forest management that can help in quantifying and monitoring changes in clearing areas. This study aimed to characterize gaps from harvested individuals, quantify the resulting forest damage, and adjust equations to describe gaps after tree cutting. Our research was conducted in three phytophysiognomies of the eastern Pará Amazon. We performed descriptive analyses using data on gap size and damage to the remaining individuals in each phytophysiognomy. We then applied predictive modeling to estimate clearing size using a generalized linear model. Modeling parameters included Gaussian, gamma, and inverse Gaussian families, with linking and transforming functions of the analyzed variables. Among the three phytophysiognomies, the largest clearings were observed in open ombrophilous forests with lianas (27,650 to 548,460 $m^2$), with 56 large gaps, 148 medium, and 113 small. The model with three linear predictors (diameter, height, and phytophysiognomy), inverse Gaussian distribution, and logarithmic link function showed the best fit. There were notable differences in clearing size across phytophysiognomies, suggesting that the phytophysiognomy should be considered when planning measures to mitigate the impacts of forest management.

**Keywords:** forest exploitation; phytophysiognomies; dense ombrophilous forest; open ombrophilous forest with lianas





## 1. Introduction

Gaps in the forest canopy are one of the consequences of logging. Cleared areas range in size from small to large and vary according to the size of the exploited tree species, the suitability of the forestry techniques [1], and the phytophysiognomy of the exploited area. Gaps created by selective exploitation favor tree regeneration due to greater light penetration and nutrient availability for the lower strata [2].

Gaps, also called forest clearings, are progressively occupied by a wide variety of species, mainly classified into three successional groups: pioneers and early and late secondary colonizers [3]. The particular dynamic of this regeneration is influenced by several factors, such as the intensity and origin of local alterations and the growth and mortality of the species occupying the clearings [4]. These factors lead to each gap being colonized by different species due to environmental conditions in and around the gap, which affect competition for light and nutrients [5,6]. The intensity of impacts in these areas depends on the size of the clearing and the degree of damage to remaining trees during extraction of the specimens selected for cutting [7–10].

Clearing size also affects the extent of the impact and colonization dynamics. It is a determinant of environmental conditions that influence plants' survival, growth, and

structuring. Specifically, clearing size influences local conditions of luminosity, temperature, humidity, and wind, among other factors [11–13]. In the Amazon rainforest, clearings are classified as large, medium, and small, and have been shown to exhibit diverse regeneration characteristics, from germination to number of specimens [14].

The Amazon, Brazil's most exploited forest, comprises several phytophysiognomies with complex and dynamic ecological processes. Therefore, any intervention in this region must be preceded by a study of its structural characteristics to achieve sustainable management [15]. These phytophysiognomies include ombrophilous forests and their variations, such as dense and open ombrophilous forests, in which environmental interactions differ considerably between these forest types.

Dense ombrophilous forests (DOFs) have upper strata comprising trees measuring 25–30 m in height with interlaced canopies. In addition, DOFs are inhabited by evergreen broadleaf trees that are typical of humid environments. Periodically flooded dense ombrophilous forests (PFDOFs) differ from DOFs because of their water regime; they experience periodic flooding due to their proximity to rivers of black or clear waters [16,17]. Open ombrophilous forests with lianas (OOFLs) have less biomass and a higher abundance of shrubs and lianas than DOFs. These features may be associated with deep water tables, impermeable soils, and poor drainage, resulting in a dense understory and an open canopy [18]. Environmental heterogeneity, which is the result of factors including topography, soil depth, temperature, and seed dispersal, is one of the main drivers of floristic composition and forest structure of forests [19].

The various characteristics of the phytophysiognomies require a different set of local approaches that may differ from those commonly used in the Amazon region. Current regulations allow timber extraction through sustainable forest management (SFM) under Normative Instruction MMA No. 5, 2006 [20], among other legal frameworks. SFM permits methods of reduced-impact exploitation (RIE), which employs scientific and engineering principles of impact mitigation, education, and training [15,21,22].

SFM, however, only considers the estimated volume to be extracted without examining the impact of logging on the remaining forest. This situation results from a scarcity of impact assessments and the absence of any legal obligation to conduct them. Thus, studies that aim to assess the impacts of factors such as clearing size are essential. Other authors have raised similar concerns, focusing on the recovery of forest stocks, assessments of the environmental and economic sustainability of forest management, and the definition of the requirements, time, and intensity of forest management [23]. Hence, the main objective of this study was to adjust generalized linear models for the prediction of clearing areas produced by logging in different phytophysiognomies of the eastern Amazon. In addition, the specific objectives were to classify and compare the clearing sizes generated by RIE between forest phytophysiognomies and evaluate the relationships between clearing size and variables such as diameter at breast height (DBH), volume and commercial height of exploited trees, and phytophysiognomy.

## 2. Material and Methods

### 2.1. Characterization of the Study Area

The study was carried out at Annual Production Unit No. 14 (APU-14) of the Forest Management Unit (FMU) at Fazenda Uberlândia (03°06′15″ 78 S; 49°53′52″ 28 W), located in Portel, Pará state. The Effective Management Area (EMA) of APU-14 is 3760.27 ha, subdivided into three areas and 39 work units (WUs) across three phytophysiognomies: DOFs, PFDOFs, and OOFLs (Figure 1 and Table 1).

The structure of DOFs, which are adapted to hot and humid climates, is characterized by a dense and interconnected canopy, an abundance of large trees, a subcanopy with individuals of medium to small size that receives little incident light, and a predominance of broadleaf species [16,17]. Similarly, PFDOF forests are rich in large trees with interconnected canopies. The difference between PFDOFs and DOFs is their water regime, since PFDOFs

are periodically waterlogged due to river floods [17]. Meanwhile, OOFLs are composed of large and isolated trees covered by lianas [17].

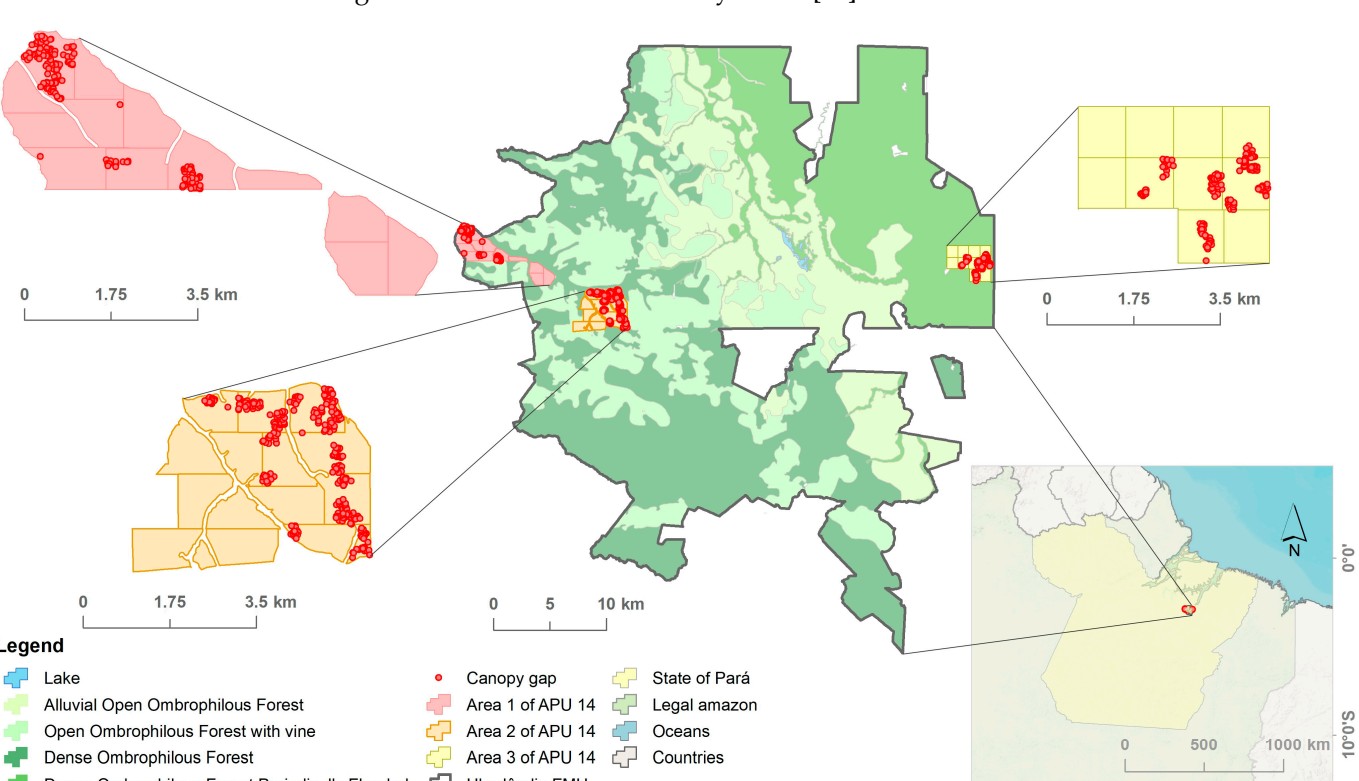

**Figure 1.** Location of the study areas and gaps and subdivisions of the Annual Production Unit No. 14 (APU-14) of the Forest Management Unit (FMU) at Fazenda Uberlândia, Portel, Pará state.

**Table 1.** Quantification of areas and phytophysiognomies in the studied Annual Production Unit.

| APU-14 | WT | TA | PPA | EFEA | Phytophysiognomy |
|--------|----|------|--------|--------|------------------|
| A1 | 11 | 1075.9 | 55.14 | 1000.6 | PFDOF |
| A2 | 14 | 1328.6 | 128.91 | 1194.5 | DOF, OOFL |
| A3 | 14 | 1356.0 | 91.47 | 1264.2 | DOF, OOFL |
| Total | 39 | 3760 | 275.53 | 3460 | - |

APU-14: Annual Production Unit-14; WT: work unit; TA: total area, in hectares; PPA: permanent preservation area, in hectares; EFEA: effective forest exploitation area, in hectares; PDFDOF: periodically flooded dense ombrophilous forest; DOF: dense ombrophilous forest; OOFL: open ombrophilous forest with lianas.

In the APU studied, all trees with a DBH ≥ 40 cm were inventoried, totaling 139 species. Identification of the phytophysiognomies was obtained using the geoprocessing tools of QGIS software. For this purpose, a vectorized classification base prepared by the company managing the APU was constructed using information from satellite images, sample collections from field trips, and manuals of Brazilian vegetation [17].

The exploitation of APU-14 by Guerra Indústria e Comércio de Madeira Ltd. was approved by the State Department for the Environment and Sustainability (Secretaria de Estado de Meio Ambiente e Sustentabilidade—SEMAS) through Authorization for Forest Exploitation AUTEF No. 273333/2019. Authorization was issued under current legislation (Normative Instruction No. 05/15—SEMAS and IBAMA Implementation Standard No.1, of 24 April 2007), which granted special rights for the exploitation of 40 commercial timber species, with the removal of 26.78 $m^3 \cdot ha^{-1}$, totaling 92,434.91 $m^3$ (13,181 trees) released for exploitation over a 35-year cutting cycle.

## 2.2. Dataset

We used data from 653 gaps caused by planned forest exploitation, representing 4.95% of the individual trees extracted in APU-14. This sample was selected to include proportional numbers of trees from each species harvested. The clearings were selected randomly from the three phytophysiognomies (n = 171 for DOF, n = 165 for PFDOF, and n = 317 for OOFL).

### 2.2.1. Determining Clearing Areas in the EFEA

Gap areas were determined by Equation (1) [24], in which *AC* = area of clearing, in m$^2$; $\pi$ = Pi value; *Bn* = largest diameter of the clearing, in meters; *Cn*: smallest diameter of the clearing, in meters (Equation (1)).

$$AC = \pi \left( \frac{B_n^2}{2} \right) \left( \frac{C_n^2}{2} \right) \tag{1}$$

Gaps were classified into three size categories: (i) small clearings, with $AC < 100$ m$^2$; (ii) medium clearings, $100$ m$^2 \leq AC < 200$ m$^2$; and (iii) large clearings, $AC \geq 200$ m$^2$.

### 2.2.2. Classification of Damage to Remaining Trees

Damage refers to the degree of injury suffered by remaining trees due to the exploitation of the trees intended for cutting. After measuring the impacted area of each selected clearing, damage was analyzed in the remaining trees with a diameter measured at 1.30 m from the ground (d1.3) greater than or equal to 10 cm. We then defined three damage intensities [25]: (i) mild, (ii) severe, and (iii) irreversible (Table 2). We counted the occurrences of each damage category by analyzing every tree damaged.

**Table 2.** Classification of damage to remaining trees.

| Damage Site | Classification According to Field Parameters | | |
| --- | --- | --- | --- |
| | **Mild** | **Severe** | **Irreversible** |
| Damage to treetops | 1/3 canopy of damaged trees | Between 1/3 and 2/3 canopy of damaged trees | Treetops destroyed |
| Damage to tree trunks | Superficial damage in the trunk of the trees | Deep damage of size < 1/2 d1.3 | Deep damage of size > 1/2 d1.3 |

In addition, to compose the forest census dataset, we compiled the following information on the extracted trees that gave rise to the clearings analyzed: (i) scientific name; (ii) d1.3 in cm; (iii) commercial height (H), in m; (iv) cross-sectional area (gi), in m$^2$; (v) volume (vi), in m$^3$; and (vi) phytophysiognomy.

## 2.3. Generalized Linear Models

The generalized linear models were calibrated using a training set based on the trees' dimensions and phytophysiognomy to predict the clearing area. The classical linear regression model assumes that the response variable has a normal distribution, constant variation, and independence. However, many situations do not meet these assumptions, so more flexible and versatile models were needed to model other functional relationships. Generalized linear models (GLMs) are one tool that has emerged to address this need [26]. GLMs expand the possible probability distributions of the dependent variable, allowing the use of distributions from the exponential families for the response variable (e.g., normal, inverse normal, gamma, binomial, and Poisson, among others).

A GLM contains three basic components: (a) a random component, (b) a systematic component, and (c) a link function [27].

(a)   Random component: this consists of the random variable Y from a set of n independent observations (y1, y2, . . . , yn) with a distribution belonging to the exponential family.

(b)   Systematic component: a linear predictor covering a set of covariates through a linear combination of parameters.

(c)   Link function: a monotonous and differentiable function that associates the random component (mean of a distribution) with the systematic component (linear predictor). Different link functions can be used for each distribution assumed for the dependent variable.

Therefore, to adjust a GLM, the behavior (distribution) of the response variable, the explanatory variable, and the link function associating the random with the systematic component must be defined. In this study, the dependent variable ($Y_i$) is the area of clearing (AC, in $m^2$) resulting from the felling of a tree, and its nature is continuous. Thus, three continuous random distributions (Gaussian, gamma, and inverse Gaussian) with the identity ($\mu$) and logarithmic (ln $\mu$) link functions and five variations of linear predictors were assumed for the conditional distribution of the response variable (Table 3).

**Table 3.** Generalized linear models adjusted to predict the area of gap (Y = *AC*) produced by the planned felling of trees in three forest phytophysiognomies of the Brazilian Amazon.

| Symbol | Linear Predictor | Family | LF |
|---|---|---|---|
| M1 | $\beta_0 + \beta_1 d + \beta_2 H + \beta_3 V$ | Gaussian | Identity |
| M2 | $\beta_0 + \beta_1 Phyto + \beta_2 d + \beta_3 H$ | Gaussian | Identity |
| M3 | $\beta_0 + \beta_1 Phyto + \beta_2 \ln(d)$ | Gaussian | Identity |
| M4 | $\beta_0 + \beta_1 Phyto + \beta_2(d^2 H)$ | Gaussian | Identity |
| M5 | $\beta_0 + \beta_1 Phyto + \beta_2 \ln(d) + \beta_3 \ln(d^2 H)$ | Gaussian | Identity |
| M6 | $\beta_0 + \beta_1 Dsev + \beta_2 Phyto + \beta_3 H$ | Gaussian | Identity |
| M7 | $\beta_0 + \beta_1 d + \beta_2 H + \beta_3 V$ | Gamma | Log |
| M8 | $\beta_0 + \beta_1 Phyto + \beta_2 d + \beta_3 H$ | Gamma | Log |
| M9 | $\beta_0 + \beta_1 Phyto + \beta_2 \ln(d)$ | Gamma | Log |
| M10 | $\beta_0 + \beta_1 Phyto + \beta_2(d^2 H)$ | Gamma | Log |
| M11 | $\beta_0 + \beta_1 Phyto + \beta_2 \ln(d) + \beta_3(d^2 H)$ | Gamma | Log |
| M12 | $\beta_0 + \beta_1 Dsev + \beta_2 Dirrev + \beta_3 Phyto + \beta_4 H$ | Gamma | Log |
| M13 | $\beta_0 + \beta_1 d + \beta_2 H + \beta_3 V$ | Inverse Gaussian | Log |
| M14 | $\beta_0 + \beta_1 Phyto + \beta_2 d + \beta_3 H$ | Inverse Gaussian | Log |
| M15 | $\beta_0 + \beta_1 Phyto + \beta_2 \ln(d)$ | Inverse Gaussian | Log |
| M16 | $\beta_0 + \beta_1 Phyto + \beta_2(d^2 H)$ | Inverse Gaussian | Log |
| M17 | $\beta_0 + \beta_1 Phyto + \beta_2 \ln(d) + \beta_3 \ln(d^2 H)$ | Inverse Gaussian | Log |
| M18 | $\beta_0 + \beta_1 Dsev + \beta_2 Phyto + \beta_3 H + \beta_4 \ln(d) + \beta_5 d$ | Inverse Gaussian | Log |

Where $\beta_0$, $\beta_1$, $\beta_3$, $\beta_4$, and $\beta_5$ = model coefficients; d = diameter at breast height (cm); H = total height (m); V = volume, in $m^3$; ln = Napierian logarithm; Phyto = phytophysiognomy; Dirrev = number of trees with irreversible damage; Dsev = number of trees with severe damage; LF = link function.

The original dataset (n = 653) was divided into training (80%) and test (20%) data through random stratified sampling based on the diameters of the trees whose extraction gave rise to the clearings. We used the training dataset to estimate the parameters of the GLMs, and the power of generalization was assessed using the test set. The fitting quality of the GLMs was assessed by inspecting the residual deviance of the Akaike information criterion (AIC) [28] (Equation (2)), the Bayesian information criterion (BIC) [29] (Equation (3)), relative root mean square error (rRMSE) (Equation (4)), and through diagnosis of half-

normal plots with simulated envelopes. The fulfillment of the hypothesis of absence of collinearity (or multicollinearity) was assessed using the variance inflation factor (VIF) statistic (Equation (5)). Severe effects of multicollinearity were assumed when VIF > 5 [30]. Potential outliers and influential points were diagnosed by analyzing a Cook's distance chart. Following this, some models were refitted by removing individual observations and evaluating the impacts of these modifications on the estimates of coefficients and standard errors.

$$AIC = -2\ln(L_P) + 2p \tag{2}$$

$$BIC = -2\ln(L_P) + K_p; \ [For \ K = \ \ln(n)] \tag{3}$$

$$rRMSE = \ \frac{100}{\overline{y}}\sqrt{\frac{1}{n}\sum_{i=1}^{n}(\hat{y}_i - y_i)^2} \tag{4}$$

$$VIF = \ \frac{1}{\left(1 - r_{23}^2\right)} \tag{5}$$

where $Lp$ = value that maximizes the maximum likelihood function of the estimated model; $p$ = number of model parameters; ln = natural logarithm; $n$ = number of observations; $y_i$ = observed value for the $i$th tree in the sample; $\overline{y}$ = observed mean of the response variable; $\hat{y}_i$ = predicted value for the $i$th tree in the sample; $r_{23}^2$ = correlation coefficient between regressors $X_1$ and $X_2$.

We performed all analyses using the R programming language, version 4.1.0. [30], adjusting all GLMs using the glm function of the *stats* package available in the R-base. We diagnosed the residuals by visual inspection of half-normal plots with simulated envelopes, using the packages auditor [31] and hnp [32].

## 3. Results

*3.1. Species Richness*

The exploited trees generating the gaps belonged to 36 species distributed in 15 botanical families. The sample sufficiency in monitoring reduced impact (5%) is commonly accepted in forest certification and implemented by companies and communities that carry out forest management. *Goupia glabra*, *Manilkara paraensis*, *Chrysophyllum venezuelanense*, and *Manilkara huberi* were the most representative species in the sample (Table 4).

Figure 2 depicts the box plots, by phytophysiognomy, for the biometric variables measured in the felled trees and the area of the clearing (response variable) formed after the felling of trees. The highest mean clearing area and clearing size variability were 141.09 m$^2$ and CV = 57.13% for OOFLs, followed by DOFs ($\overline{AC}$ = 101.48 m$^2$, CV = 49.8%) and PFDOFs ($\overline{AC}$ = 82.8 m, CV = 42.9%).

The mean diameters of trees from the OOFLs ($\overline{d}_{1,3}$ = 77 cm) and DOFs ($\overline{d}_{1,3}$ = 76.8 cm) were similar, with a dispersion of less than 26%. In PFDOFs, the trees showed lower variability (CV = 20.1%) and mean diameter ($\overline{d}_{1,3}$ = 69.2 cm). Average tree height was higher and less variable in DOFs ($\overline{H}$ = 18.46 m, CV = 10.7%) than in the other phytophysiognomies. In PFDOFs, average tree height was the lowest ($\overline{H}$= 13.9 m), though with high variance (CV = 19.5%). Finally, the highest average wood volume of the trees was measured in DOFs ($\overline{V}$ = 6.30 m$^3$), followed by OOFLs ($\overline{V}$ = 5.64 m$^3$) and PFDOFs ($\overline{V}$ = 3.87 m$^3$). The variance of the variable volume was high, above 50% for all phytophysiognomies.

**Table 4.** Number of trees, arranged by species and phytophysiognomy, related to gap formation after extraction by reduced impact exploitation methods.

| Species | Family | DOF | PFDOF | OOFL | Total |
|---|---|---|---|---|---|
| *Astronium lecointei* Ducke | Anacardiaceae | 2 | - | 10 | 12 |
| *Caryocar glabrum* (Aubl.) Pers. | Caryocaraceae | - | 9 | 1 | 10 |
| *Caryocar villosum* (Aubl.) Pers. | Caryocaraceae | 1 | - | 2 | 3 |
| *Chrysophyllum venezuelanense* (Pierre) T.D.Penn. | Sapotaceae | 11 | 2 | 49 | 62 |
| *Cordia goeldiana* Huber | Boraginaceae | 2 | - | - | 2 |
| *Couratari guianensis* Aubl. | Lecythidaceae | 4 | - | 32 | 36 |
| *Couratari stellata* A.C.Sm. | Lecythidaceae | 1 | 1 | 3 | 5 |
| *Dinizia excelsa* Ducke | Fabaceae | 15 | - | 20 | 35 |
| *Diplotropis martiusii* Benth. | Fabaceae | 2 | 24 | - | 26 |
| *Diplotropis purpurea* (Rich.) Amshoff | Fabaceae | 2 | 5 | 2 | 9 |
| *Dipteryx polyphylla* Huber | Fabaceae | 8 | 13 | 8 | 29 |
| *Endopleura uchi* (Huber) Cuatrec. | Humiriaceae | 1 | - | 2 | 3 |
| *Enterolobium schomburgkii* (Benth.) Benth. | Fabaceae | 4 | - | 3 | 7 |
| *Erisma uncinatum* Warm. | Vochysiaceae | - | 1 | - | 1 |
| *Goupia glabra* Aubl. | Celastraceae | 10 | 48 | 22 | 80 |
| *Hymenaea courbaril* L. | Fabaceae | 5 | 11 | 3 | 19 |
| *Hymenolobium petraeum* Ducke. | Fabaceae | 5 | 11 | 4 | 20 |
| *Iryanthera paraensis* Huber | Myristicaceae | - | 5 | 4 | 9 |
| *Lecythis pisonis* Cambess. | Lecythidaceae | 1 | - | 1 | 2 |
| *Licaria cannella* (Meisn.) Kosterm. | Lauraceae | 4 | - | 3 | 7 |
| *Manilkara huberi* (Ducke) Chevalier | Sapotaceae | 21 | - | 38 | 59 |
| *Manilkara paraensis* (Huber) Standl. | Sapotaceae | 36 | 4 | 22 | 62 |
| *Micropholis venulosa* (Mart. & Eichler) Pierre | Sapotaceae | 7 | - | 12 | 19 |
| *Ocotea neesiana* (Miq.) Kosterm. | Lauraceae | - | - | 2 | 2 |
| *Ocotea rubra* Mez | Lauraceae | 4 | - | 7 | 11 |
| *Piptadenia suaveolens* Miq. | Fabaceae | 3 | - | 15 | 18 |
| *Pouteria oblanceolata* Pires | Sapotaceae | 8 | - | 4 | 12 |
| *Qualea paraensis* Ducke | Vochysiaceae | - | 6 | - | 6 |
| *Sclerolobium paraense* Huber | Fabaceae | 2 | - | - | 2 |
| *Simarouba amara* Aubl. | Simaroubaceae | - | 6 | 5 | 11 |
| *Sterculia alata* Roxb. | Malvaceae | 3 | 3 | 3 | 9 |
| *Terminalia amazonica* (J.F.Gmel) Exell. | Arecaceae | 1 | - | 11 | 12 |
| *Tetragastris panamensis* (Engl.) Kuntze | Burseraceae | | - | 18 | 18 |
| *Vantanea parviflora* Lam. | Humiriaceae | 6 | - | 3 | 9 |
| *Vatairea paraensis* Ducke | Sapotaceae | 2 | 2 | 8 | 12 |
| *Vochysia guianensis* Aubl. | Vochysiaceae | | 14 | - | 14 |
| Total | 36　　　15 | 171 | 165 | 317 | 653 |

PFDOF: periodically flooded dense ombrophilous forest; DOF: dense ombrophilous forest; OOFL: open ombrophilous forest with lianas.

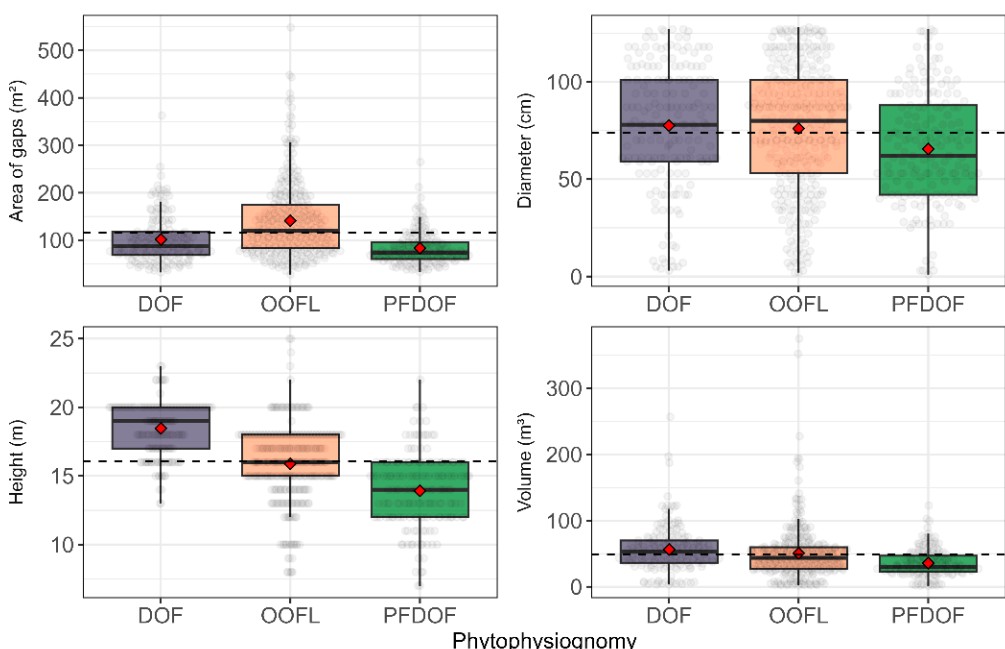

**Figure 2.** Box plot for the area of gaps (dependent variable) and the dendrometric variables measured in the felled trees, arranged by phytophysiognomy. Vertical bars (black color): Q1 − 1.5 × IQR (1st quartile minus 1.5 times the interquartile range) and Q3 + 1.5 × IQR (3rd quartile plus 1.5 times the interquartile range); red points: arithmetic mean of the variables of each phytophysiognomies; horizontal dashed lines (black color) in each sub-\figure: arithmetic mean of the variable in the sample. PFDOF: periodically flooded dense ombrophilous forest; DOF: dense ombrophilous forest; OOFL: open ombrophilous forest with lianas.

### 3.2. Dimensioning of the Clearings by Phytophysiognomies

Of the sampled clearings, 342 were classified as small, 244 as medium, and 67 as large. In relative terms, felled trees generated a higher proportion of medium (*n* = 148; 46.7%) and large (*n* = 56; 17.7%) clearings in the OOFLs than in the other forest types. The medium and large clearings represented 46.7% (20,900 m$^2$) and 34.9% (15,622 m$^2$), respectively, of the total area opened by the felled trees. The highest proportions of small clearings were found in DOF (*n* = 101; 59.1%) and PFDOF (*n* = 128; 77.6%) phytophysiognomies. Thus, small clearings were 40.4% (7002 m$^2$) of the open area in DOFs and 63.3% (8653 m$^2$) in PFDOFs (Figure 3).

The large clearings in the OOFLs were generated by the felling of trees of relatively small diameter (50 cm), and the number of damaged trees during the formation of each of these gaps ranged from 10 to 30. In DOF, large clearings were generated from the felling of trees with diameters above 75 cm, with 10 to 30 damaged trees per clearing. These gaps in PFDOFs resulted from cutting trees above 100 cm in diameter, with damaged trees numbering 10 to 30.

The trees whose felling gave rise to small and medium clearings in the OOFLs and DOFs had heights below 12 m, and the number of damaged specimens ranged from 0 to 10. In comparison, large clearings were caused by trees of over 12 m in height, with 10 to 30 damaged specimens. In PFDOFs, meanwhile, large gaps were found in areas where the exploited trees were over 14 m in height, with 10 to 20 damaged specimens.

The timber volume of trees whose felling caused large clearings was around 1.5 m$^3$ in OOFLs, with 0 to 30 damaged specimens. In DOF, the large clearings were formed from the felling of trees above 3 m$^3$, with 10 to 30 damaged specimens. Large gaps in PFDOFs were generated after exploiting trees above 10 m$^3$, with 0 to 20 damaged specimens.

OOFLs showed the highest number and percentage of damaged trees in the three categories (mild, severe, and irreversible), followed by DOFs and PFDOFs (Figure 4).

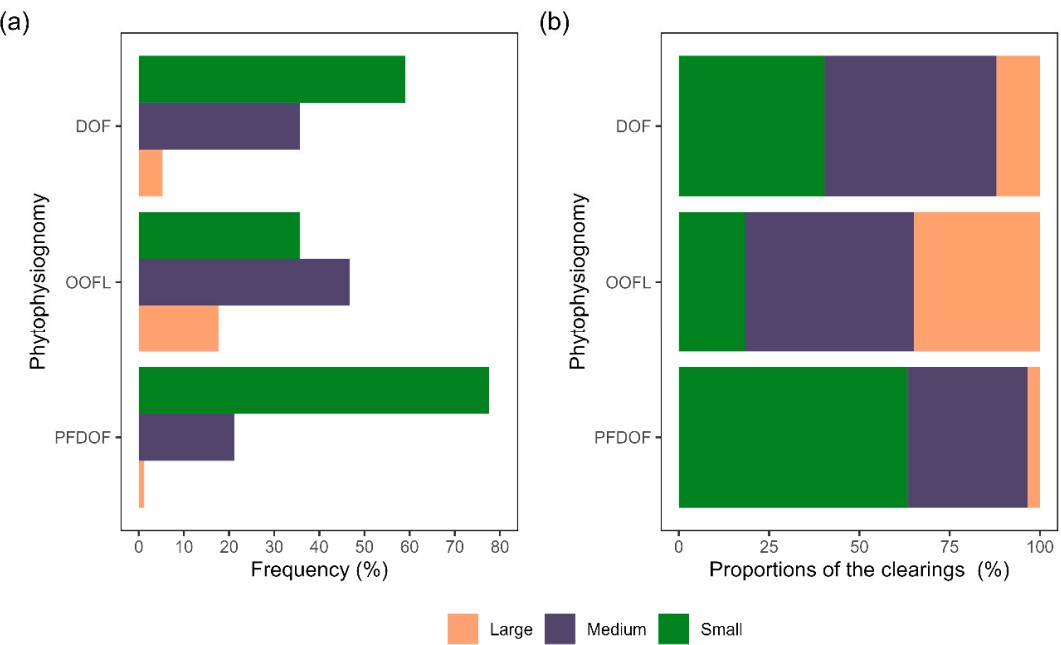

**Figure 3.** Frequency (**a**) and proportions (**b**) of the gaps caused by the felling of the trees by directional cutting of reduced impact on the studied phytophysiognomies. The highest frequencies (**a**) and proportions (**b**) of large gaps are found in the OOFL phytophysiognomy, and of small gaps in PFDOF. PFDOF: periodically flooded dense ombrophilous forest; DOF: dense ombrophilous forest; OOFL: open ombrophilous forest with lianas. Small: clearing < 100 m$^2$; medium: 100 m$^2$ ≤ clearing < 200 m$^2$; large: clearing ≥ 200 m$^2$.

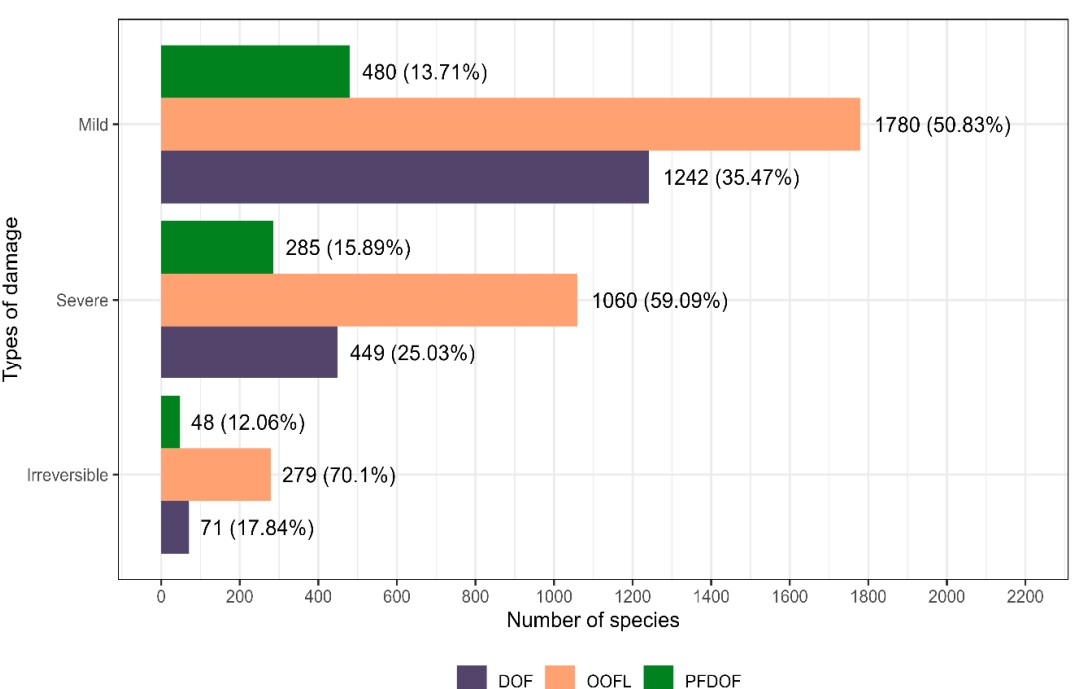

**Figure 4.** Proportion and number of mild, severe, and irreversible damages by phytophysiognomy. PFDOF: periodically flooded dense ombrophilous forest; DOF: dense ombrophilous forest; OOFL: open ombrophilous forest with lianas.

*3.3. Generalized Linear Models*

Table 5 describes the estimates of the regression coefficients and their significance through the *t*-test (α = 0.05) and the adjustment statistics of the models.

Most of the calibrated models are unreliable for making inferences since they had nonsignificant regression coefficients ($\alpha$ = 0.05; *t*-test). Severe multicollinearity effects (VIF > 5) were identified in the coefficient estimates of the fitted models that included the inverse Gaussian (logarithmic link) distribution and also for the models M5 (Gaussian and identity link) and M11 (gamma and logarithmic link), which include the predictors ln(d) and ln(d²H). The M12 model (gamma and logarithmic link), with the inclusion of predictors indicative of the amount and intensity of damage caused by falling trees on remaining trees, fitted the data better and showed better predictive performance. In this case, predictors indicative of the amount of irreversible or severe damage to trees were more important in describing the effects on the response variable. For the M12 model, the half-normal plots showed residuals inside the simulated envelope, indicating a good fit of the model to the data (solid black lines, Figure 5). In addition, the randomized quantile residuals showed good adherence to the Gaussian distribution (Figure 6), confirmed by the Shapiro–Wilk hypothesis test, without discrepant values, which were in the range of −3 to 3.

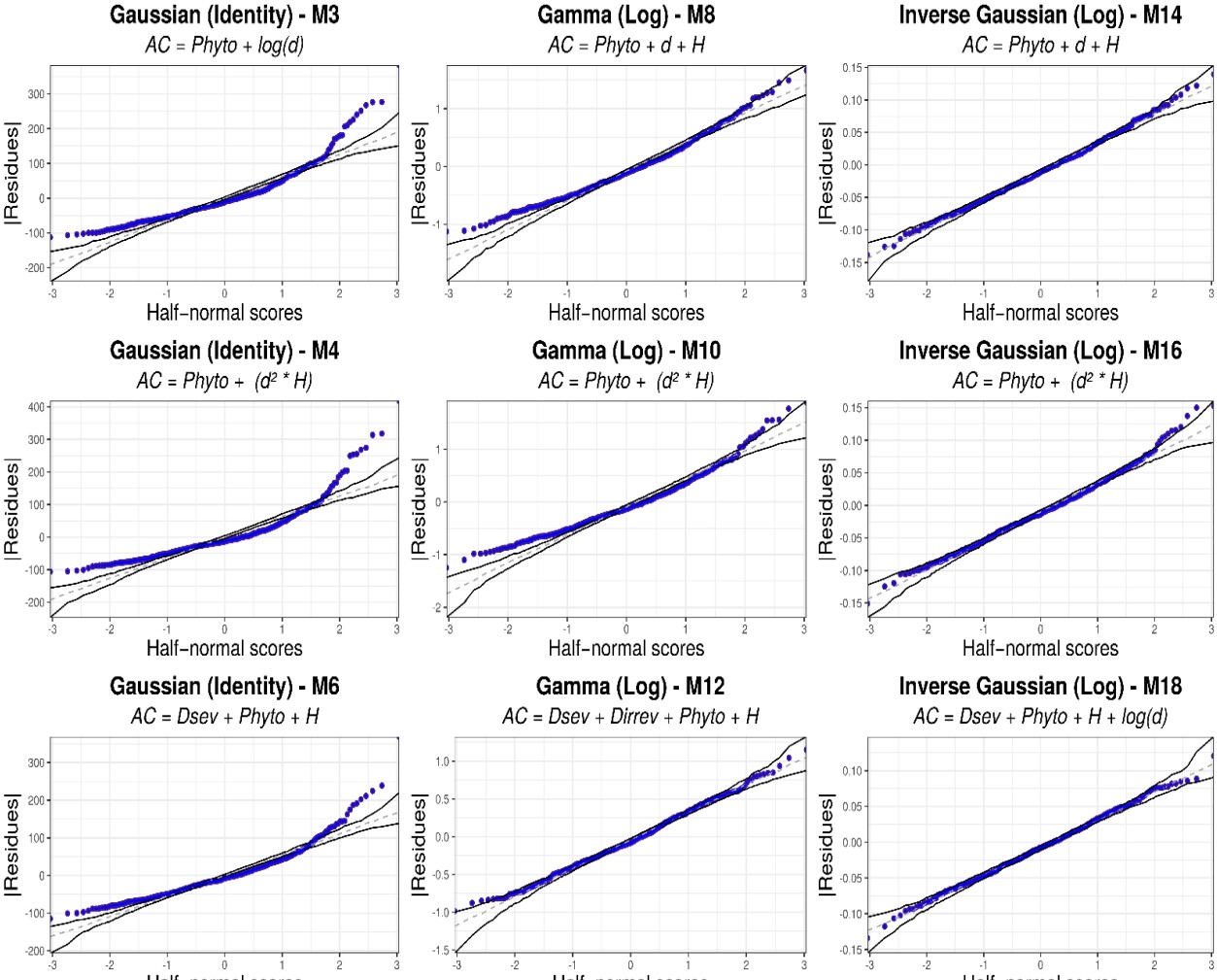

**Figure 5.** Half-normal plots with simulated envelope for the deviation residuals of the six models with the best fitting. Predictors indicative of the amount of irreversible or severe damage to trees were more important in describing the effects on the response variable. For the M12 model, the half-normal plots showed residuals inside the simulated enve-lope, indicating a good fit of the model to the data. *AC* = area of the clearing; *Phyto* = phytophysiognomy; *d* = diameter at breast height; *H* = height; *Dirrev* = number of trees with irreversible damage; *Dsev* = number of trees with severe damage; log = logarithmic transformation.

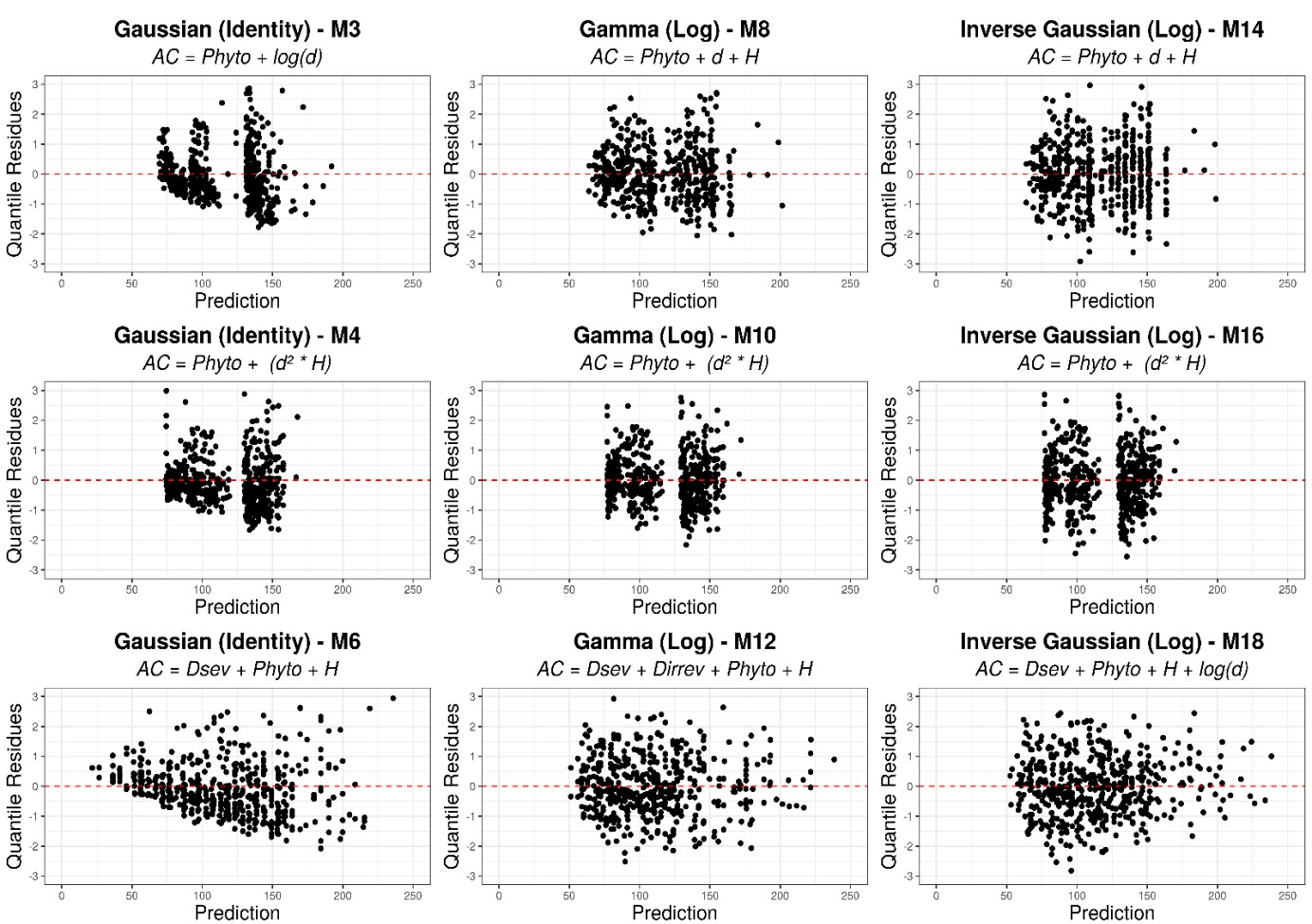

**Figure 6.** Randomized quantile residue plots of the six best-fit models. *AC* = area of the clearing; *Phyto* = phytophysiognomy; *d* = diameter at breast height; *H* = height; *Dirrev* = number of trees with irreversible damage; *Dsev* = number of trees with severe damage; log = logarithmic transformation.

**Table 5.** Coefficients and precision statistics of generalized linear models to predict the clearing size in an area of planned forest exploitation in the Brazilian Amazon.

| Mod | Coefficients | | | | | | Adjustment Statistics | | | | | |
|-----|-----|-----|-----|-----|-----|-----|-----|-----|-----|-----|-----|-----|
| | | | | | | | Set Training | | | Set Test | | |
| | $\beta_0$ | $\beta_1$ | $\beta_2$ | $\beta_3$ | $\beta_4$ | $\beta_5$ | AIC | BIC | rRMSE | AIC | BIC | rRMSE |
| M1 | −30.0983 | 1.525 * | 1.7572 | 0.5655 | - | - | 5838.92 | 5860.24 | 56.26 | 5881.24 | 5902.55 | 55.24 |
| M2 | −11.7493 | −44.9291 * | 44.1305 * | 1.4245 * | 2.7063 * | - | 5771.37 | 5796.95 | 54.46 | 5617.29 | 5638.6 | 53.25 |
| M3 | −402.316 * | −38.476 * | −47.945 * | 125.77 * | - | - | 5776.53 | 5797.85 | 54.75 | 5565.34 | 5586.65 | 56.14 |
| M4 | 91.56 * | −37.38 * | −49.98 * | 0.0079 * | - | - | 5784.94 | 5806.25 | 55.09 | 5849.14 | 5874.72 | 53.32 |
| M5 | −476.198 * | −44.977 * | −43.689 * | 38.683 | 39.528 * | - | 5771.66 | 5797.24 | 53.52 | 5555.88 | 5581.47 | 54.47 |
| M6 | −5.859 | 20.411 * | −36.245 * | −16.484 * | 4.899 * | - | 5708.05 | 5733.63 | 47.62 | 5708.05 | 5733.63 | 47.56 |
| M7 | 3.1836 * | 0.0169 * | 0.025 * | −0.0263 | - | - | 5565.97 | 5587.29 | 57.25 | 5505.97 | 5531.55 | 57.59 |
| M8 | 3.7453 * | −0.3555 * | −0.4278 * | 0.011 * | 0.0204 * | - | 5467.08 | 5492.66 | 54.36 | 5852.67 | 5873.98 | 53.03 |
| M9 | 0.6113 | −0.3118 * | −0.4542 * | 0.9984 * | - | - | 5466.61 | 5487.93 | 54.62 | 5565.48 | 5586.8 | 55.49 |
| M10 | 4.565 * | −0.3495 * | −0.438 * | 0.0000035 * | - | - | 5478.48 | 5499.79 | 55.03 | 5516.7 | 5538.02 | 53.32 |
| M11 | 0.093 | −0.357 * | −0.4235 * | 0.3136 | 0.3051 * | - | 5461.15 | 5486.73 | 53.22 | 5862.59 | 5883.91 | 53.69 |
| M12 | 3.567 * | 0.1162 * | 0.2111 * | −0.2289 * | −0.095 * | 0.0461 * | 5298.14 | 5327.98 | 41.22 | 5298.14 | 5327.98 | 47.64 |
| M13 | 3.4579 * | 0.0134 * | 0.0154 | 0.0024 | - | - | 5521.60 | 5542.92 | 57.36 | 5569.44 | 5590.76 | 57.47 |

**Table 5.** *Cont.*

| Mod | Coefficients | | | | | | Adjustment Statistics | | | | | |
|---|---|---|---|---|---|---|---|---|---|---|---|---|
| | | | | | | | Set Training | | | Set Test | | |
| | $\beta_0$ | $\beta_1$ | $\beta_2$ | $\beta_3$ | $\beta_4$ | $\beta_5$ | AIC | BIC | rRMSE | AIC | BIC | rRMSE |
| M14 | 3.7237 * | −0.3407 * | −0.4372 * | 0.0122 * | 0.0166 * | - | 5424.48 | 5450.06 | 54.39 | 5517.69 | 5539.01 | 52.93 |
| M15 | 0.521 | −0.3096 * | −0.4582 * | 0.0198 * | - | - | 5423.73 | 5445.05 | 54.76 | 5830.74 | 5856.32 | 55.31 |
| M16 | 4.48 * | −0.3459 * | −0.4309 * | 0.0000044* | - | - | 5428.13 | 5449.44 | 55.03 | 5537.03 | 5562.61 | 53.38 |
| M17 | 0.1709 | −0.3456 * | −0.4354 * | 0.4295 | 0.2544 * | - | 5420.80 | 5446.38 | 53.41 | 5491.88 | 5517.46 | 53.48 |
| M18 | 3.9067 * | 0.1456 * | −0.2808 * | −0.1968 * | 0.0375 * | −0.0244 | 5371.06 | 5400.91 | 46.01 | 5371.06 | 5400.91 | 47.70 |

Mod = model; $\beta_0$, $\beta_1$, $\beta_2$, $\beta_3$, $\beta_4$, and $\beta_5$ = model coefficients; AIC = Akaike information criterion; BIC = Bayesian Schwarz criterion; rRMSE = relative root mean square error; * = significance level ($\alpha$ = 0.05).

## 4. Discussion

### 4.1. Species Richness

The species with the highest number of individuals sampled in the clearings follow the pattern of forest management in the Amazon region in terms of both abundance of species and commercial demand. Specifically, it is consistent with a report of the 10 most managed species of the lower Amazon in a study conducted between 2006 and 2016. That study reported more than 4 million m$^3$ of log extraction, with emphasis on the species *Manilkara huberi* (Ducke) Chevalier and *Goupia glabra* Aubl. [33]. Studies conducted by the authors of [34,35] also addressed the abundance and commercial interest of these species in management areas located in other regions, showing that a specific logging profile existed throughout the Amazon that placed significant pressure on the most exploited species. Hence, the selective exploitation of forest species in the Amazon causes unequal pressure among flora populations, restructuring the forest with several plant species different from those found in the original forest [36].

The flora populations of each phytophysiognomy exhibit distinct features. One of these features is tree density, since DOFs and PFDOFs have a higher number of individuals per hectare than OOFLs. A study carried out in another DOF reported a density of 544 trees per hectare ($\geq$10 cm), while a study carried out in an OOFL reported 306 individuals per hectare [37]. These studies demonstrate that the number of specimens around the exploited trees differs between phytophysiognomies.

Other characteristics that differ between DOFs, PFDOFs, and OOFLs are flooding regime, liana density, and tree size and diameter [38–40]. These characteristics are not considered during forest management activities since the legal framework requires that different forest areas be treated with the same silvicultural and exploitation techniques. Thus, there is no differential logging for OOFLs, which are areas with more lianas per hectare and thicker and taller specimens than in the other phytophysiognomies. Consequently, logging activities generate more extensive clearings with more damaged remaining trees in OOFLs than in the other ombrophilous forests. The generalization of forest management techniques for all tropical forests—or even all ombrophilous forests—therefore leads to more extensive damage to trees in clearings of OOFLs than in DOFs and PFDOFs.

### 4.2. Measuring the Clearings by Phytophysiognomy

Although the harvested trees in DOFs are greater in height and average volume than in PFDOFs, there were more damaged trees in clearings of PFDOFs than in DOFs. This outcome is probably due to the more pronounced occurrence of lianas in PFDOFs than in DOFs. Other studies carried out in Amazonian Forest areas with a prominent presence of lianas demonstrate relationships between the density of lianas, the clearing size, and damage to remaining trees [41]. It should be noted that an OOFL has few large trees per hectare, its canopies are connected by lianas, and there is high incident light. In contrast, dense forests have many large specimens per hectare and dense, connected canopies [17]. Thus, the presence of lianas connecting the trees is one of the reasons for the opening of more extensive clearings and more damaged trees in OOFLs than in the other ombrophilous

forests [42]. More extensive clearings are a problem for forest regeneration since they are avoided by fauna due to difficulties in their structural characteristics, rendering the dispersal of fruits and seeds more difficult [43]. In addition, reports indicate that clearings are affected by increased temperature and loss of humidity since they are exposed to high luminosity [4,9].

Hence, forest management regulations need to consider the characteristics of forest phytophysiognomies when considering exploitation intensity, liana-cutting level, and specific silvicultural treatments, among other factors. Forest management regulations currently dictate that lianas must be cut one year before forest exploitation in all forest types [17,20,44]. However, this criterion could be revised by inserting new guidelines for tailoring reduced impact management to each phytophysiognomy and considering factors such as tree density (specimens per hectare) and degree of liana occurrence.

The high density of specimens over 10 cm around the exploited trees, associated with the presence of lianas, increases the damage caused by forest exploitation on remaining trees, both in the number of affected specimens and the severity of the impact. Consequently, the size of the clearings also increases, which may result in the establishment of pioneer species with low timber value [45].

After examining the relationships between the morphometric variables across these three forest typologies, it appears that there are large clearings with a high number of damaged specimens in the analyzed OOFL, which showed the smallest DBH, height, and volume. Since they can also generate large clearings, other variables to consider are the type of canopy, silvicultural treatments, presence of lianas, density [13,37,46], water deficit, soil fertility, flooding, and forest degradation [47].

There was a higher percentage of individual trees with mild, severe, and irreversible damage in PFDOFs than in DOFs and OOFLs. Thus, PFDOF areas require different silvicultural treatments than those used in other phytophysiognomies, specifically to remove the lianas attached to the trees to be cut [38]. The greater the number of liana connections between the canopy of a tree and the surrounding trees, the greater the amount and degree of damage to the remaining trees [48]. Cutting lianas six months before logging would significantly reduce the damage to the remaining trees. Low-impact management areas currently conduct liana cuts one year before exploitation during the forest inventory.

The gaps opened during logging are little studied, and there have been no previous studies on the impact of planned forest management in the exploited area. Therefore, the prediction of clearing size could be made a mandatory requirement before exploitation using existing models for the first APU and adjusted models on subsequent APUs. The utility of this requirement relies on the fact that the affected area is fully linked to forest restructuring to allow returning for another exploration cycle via forest management.

In forest management, Brazilian law allows re-exploitation after 25 to 35 years, with the volume to be exploited in a second cutting cycle dependent upon the impact caused to the trees that remained after the first cutting cycle [20]. Many studies show that exploited areas do not reach their pre-exploitation volume during this period, seeing only a return of about 30%–40% of their previous volume, along with major changes in species composition after recolonization [49–51]. Forest exploitation in tropical environments requires species classification according to their ecophysiological demands and the establishment of a cutting cycle compatible with the recovery of the volume extracted from each group, ensuring the environmental and financial sustainability of logging activity [23].

*4.3. Generalized Linear Models*

Most studies use linear regression models to predict the size of clearings from variables such as DBH and the volume and morphometry of the canopy. In a previous study [46], dendrometric variables such as DBH, basal area, and volume, along with morphometric variables of the canopy such as its length, average diameter, branch length, and projection area, were used to predict clearing areas in forest management units of the state of Acre. Other authors used these same variables to obtain accurate estimates in five forest manage-

ment areas in the Mamirauá Sustainable Development Reserve [52]. None of these studies considered phytophysiognomy as a variable in their adjusted equations, which highlights the importance of the present study.

This study showed a significant difference in the size of the clearings between phytophysiognomies. Hence, using phytophysiognomy as a variable in models predicting clearing size would be a way to monitor and reduce the areas impacted by forest management. However, complementary long-term studies on the relationship between clearing size, log volume recovery, forest structure, and diversity of the exploited area in different phytophysiognomies may also be informative.

Studies indicate a highly dynamic generation of clearings in the western and eastern Amazon region, with some specific cases in the central Amazon [47]. This concerning finding demonstrates the need for forestry operations to focus on reducing clearing size.

The difficulty in using standard analyses such as regression for predicting clearings in tropical forests is due to the assumption of normal distributions. Therefore, GLMs are more suitable since they allow data analyses without this assumption [53]. Among all the tested and adjusted models that had a 5% level of statistical significance, those of the inverse Gaussian family showed the best fit to the data, a result consistent with the positive asymmetric organization of the database [54,55]. Among these models, the M12 stands out with the best fit to the database, the lowest AIC and BIC values, and better distribution of randomized residuals. Thus, the M12 model is feasible and easy to apply to the reality of Amazon Forest managers since it uses data already collected during forest management. Some of these data are DBH and height, collected during the 100% inventory, along with the phytophysiognomy classification, which is conducted when completing the annual operational plan.

## 5. Conclusions

The felling of trees has opened large clearings in all the studied phytophysiognomies. However, the clearings formed are smaller in diameter and volume in OOFLs than in DOFs and PFDOFs.

The floristic units differed among the studied clearings, indicating the need for management prerogatives that take the phytophysiognomy of the exploited area into account. Features related to the phytophysiognomy, such as tree density, DBH, height, and volume, density of lianas, and water regime, can drastically interfere with forest management. Management that does not consider these features can cause more damage to the remaining forest, which can hinder forest recovery and re-exploitation of the same area according to the cycle established in current laws and standards.

The size of the clearings is an essential variable for monitoring forest management since it is linked to the proportion of impact on the forest and its regeneration capacity.

The easy-to-apply M12 model showed the best fit for the database, demonstrating that phytophysiognomy is a significant variable in predicting clearing size. Thus, forest management should consider phytophysiognomy for the specific implementation of silvicultural measures intended to minimize impact on the forest.

**Author Contributions:** Data curation, A.S.d.S.; formal analysis, S.L.Z., D.V.S. and A.S.d.S.; investigation, S.L.Z. and N.C.G.; methodology, D.V.S.; project administration, E.J.H.R.; supervision, E.J.H.R.; writing—original draft, S.L.Z.; writing—review and editing, N.C.G., D.V.S., E.J.H.R. and S.D.B.d.S. All authors have read and agreed to the published version of the manuscript.

**Funding:** This study was supported by the Federal University of Pará-UFPA, through the Qualified publication support program-PAPQ for paying the publication fee. (Process 23073.022878/2023-42).

**Data Availability Statement:** Not applicable.

**Conflicts of Interest:** The authors declare no conflict of interest.

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
