# Peer review of "Generalized Linear Models for Describing Tree Gaps in Forest Management Areas in the Brazilian Amazon"

_forests, doi:10.3390/f14040841_

Round 1

Reviewer 1 Report

Thanks for giving me a chance to be a referee to review this manuscript. I am fond of Generalised mixed models and tropical forest change dynamics. Overall, this is a rigorously designed study with a significant synthesis. I like the idea of applying the generalized linear models to quantify the size f clearing areas and forest cover change in the Brazilian Amazon.  The approach is efficient and the inferences derived from the study meet the aimed scope of the study. I evaluate this as a well-written manuscript with rare places that deserve major changes. I have some gross comments and detailed suggestions that need authors to pay attention to and modify to improve the quality of the study. I ask the authors to modify the English in the article because I find some expressions of sentences may be not so native for an international readership. Thereafter, I may accept the submission for publication.

Title: Very good, very clear, but a bit long.

Abstract: Sounds good, authors may consider adding quantitative results. What is M12?

Introduction

The introduction is very good; the authors demonstrate a thorough knowledge of the published literature and highlight the importance and background to carry out this investigation.

 Materials and Methods

Figure 1 is good.

Table 1 Typo in the first row and first column

Methods are technically strong and well explained.

Results

The results are clear and well-presented.

Figure 2. The y-axis could be added for each graph with measuring units.

Figure 5. High-resolution image?

Discussion

Good.

Conclusion: No comments

Reviewer 2 Report

In this work some generalized linear models were calibrated using ground data (H,V, d1,3 etc.) to estimate clearing areas size.

The work is well structured, but an extensive language editing is required. Many unusual sentences/works are present. For example, L165 “generalized linear models were adjusted using a training set”. Models are ordinarily calibrated not adjusted. Moreover, Table 5, “set test” or “set training”. They are “training/test sets”. Concerning the experimental design, authors did not explore the collinearity among predictors in proposed GLMs. I suppose that d1,3 and H and V are intercorrelated therefore model’s parameters are deeply affected by this phenomenon inflating general deduction proposed by authors. Furthermore, authors never mentioned some error metrics like R2 or MAE or RMSE but only relative model performance metric like AIC or BIC. Without error metrics is impossible to critically asses the accuracy / reliability of proposed models, making the entire work  not supported by results and pure speculative.  Another problem concerns the damage level parameter. Why was it not used in GLM? There is no linking between GLM and damages analysis. A deeper explanation about their joint adoption in this work is needed.

Finally, I suggest a major revision of current manuscript.

Round 2

Reviewer 2 Report

The authors have adressed to all my comments. Therefore, I believe that the manuscript is now ready to be published.